# A High-Linearity Glucose Sensor Based on Silver-Doped Con A Hydrogel and Laser Direct Writing

**DOI:** 10.3390/polym15061423

**Published:** 2023-03-13

**Authors:** Yulin Hu, Dasheng Yang, Hongbo Zhang, Yang Gao, Wenjun Zhang, Ruixue Yin

**Affiliations:** 1School of Mechanical and Power Engineering, East China University of Science and Technology, 130 Meilong Road, Shanghai 200237, China; 2Shanghai Key Laboratory of Intelligent Sensing and Detection Technology, East China University of Science and Technology, 130 Meilong Road, Shanghai 200237, China; 3Division of Biomedical Engineering, University of Saskatchewan, 57 Campus Drive, Saskatoon, SK S7N 5A9, Canada

**Keywords:** laser direct writing, enzyme-free glucose sensor, silver particles, glucose-responsive, hydrogels

## Abstract

A continuous glucose monitoring (CGM) system is an ideal monitoring system for the blood glucose control of diabetic patients. The development of flexible glucose sensors with good glucose-responsive ability and high linearity within a large detection range is still challenging in the field of continuous glucose detection. A silver-doped Concanavalin A (Con A)-based hydrogel sensor is proposed to address the above issues. The proposed flexible enzyme-free glucose sensor was prepared by combining Con-A-based glucose-responsive hydrogels with green-synthetic silver particles on laser direct-writing graphene electrodes. The experimental results showed that in a glucose concentration range of 0–30 mM, the proposed sensor is capable of measuring the glucose level in a repeatable and reversible manner, showing a sensitivity of 150.12 Ω/mM with high linearity of R^2^ = 0.97. Due to its high performance and simple manufacturing process, the proposed glucose sensor is excellent among existing enzyme-free glucose sensors. It has good potential in the development of CGM devices.

## 1. Introduction

The incidence of diabetes has gradually increased in recent decades and diabetes has become a chronic non-communicable disease that can seriously threaten human health. According to the data of the International Diabetes Mellitus Federation (IDF), about 463 million people worldwide had diabetes in 2019, i.e., 9.3% of adults aged 20–79 years had diabetes [1]. Diabetes now has become a public health problem globally. Diabetes mellitus is a chronic metabolic disease caused by impaired insulin biological function or defective secretion of insulin. It is mainly classified as type I diabetes and type II diabetes. Patients with both types of diabetes can suffer from serious complications, including kidney failure, blindness, and foot and leg damage, and it may even lead to amputation, heart attack, and stroke [2]. In order to improve the curative effect of treatment, control insulin levels, and reduce the mortality rate of diabetes, the artificial closed-loop system, namely the artificial pancreas system, is widely studied by researchers. Traditional glucose detection equipment is mainly based on the electrochemical method. Frequent analysis of blood glucose (BG) levels requires obtaining a small blood sample (<1 µL) via the “finger prick” collection method, which is inconvenient and results in poor patient compliance. In addition, this method cannot monitor the glucose level of patients in real time, which will increase the incidence of patients. Accurate and convenient human glucose detection is an effective method for the early diagnosis of diabetes and prevention of complications [3].

Glucose biosensors are mainly used in biochemistry, clinical chemistry, and food analysis for glucose quantitative detection requirements and in clinical medicine for blood glucose detection requirements. The electrochemical sensors are categorized into enzymatic and non-enzymatic types [4]. In 1962, Clack and Lyons [5] developed the first enzyme electrode whose surface is modified with a glucose-sensitive enzyme which reduces oxygen to hydrogen peroxide. In 1967, the glucose oxidase (GOD) electrode was developed by Updik et al. to measure blood glucose [6]. With the continuous progress of the sensor based on glucose oxidase, the measurement accuracy has been improved. However, the maintenance of enzymatic activity, the stability of immobilized enzymes, and the long-term stability may hinder further development of enzyme-based sensors for continuous glucose monitoring [7]. The longest life cycle of enzyme-based continuous glucose sensors for both clinic and research has been reported as 14 days [8]. In subsequent studies, Ag, Pt, Au, and their oxides CuO, NiO, MnO, or ZnO were used as catalysts to catalyze glucose to make glucose sensors [9,10]. Lee et al. [11] used patch-type Pt black-coated stainless steel/Au microneedles for minimally invasive continuous glucose monitoring. The Pt black microneedle showed wide linear range (0.05–36 mM), good anti-interference, and good stability (6 days) in 0.1 M PBS (pH 7.4). You et al. [12] reported the superior catalytic property of 4.5% CuO/Cu(OH)_2_ compared with 2.6% for copper oxide/hydroxide nanoparticles with carbon film through co-sputtering copper and carbon at the same time. The results indicate that a higher amount of CuO/Cu(OH)_2_ is responsible for a higher electro-oxidation sensitivity toward glucose. Wooten et al. [13] proposed a comparable study between conventional Au disk electrodes and Au nanostructured film-modified electrodes. It was found that the conventional electrodes provided similar electroanalytical benefits while requiring much simpler and shorter preparation as compared to more complex nanostructured gold electrodes with only limited electroanalytical benefits as obtained from amperometric sensors for glucose in physiological conditions. Due to the ease of reaction of metals and their oxides with active molecules, lack of glucose selectivity, and high manufacturing cost, glucose-sensitive materials have been investigated as another possibility for non-enzyme glucose sensing.

In clinical testing, glucose sensors are integrated into the continuous glucose monitoring system (CGMS) for real-time glucose data collection from patients. CGM can overcome the inconvenience caused by fingertip blood collection and the disadvantage of not being able to track blood glucose fluctuations [14,15] and is used to help patients with diabetes and their healthcare providers more effectively manage care [16]; so, the development of CGM devices has received increasing attention [17,18]. They provide important information for diabetic control, including the instantaneous real-time display of the glucose level and glucose change rate, alerts and alarms for actual or impending hypo- and hyperglycemia, “24/7” coverage, and the ability to characterize glycemic variability, so the preparation of flexible glucose sensors with excellent performance is the focus of the development of next-generation CGM devices. Until now, the development of flexible glucose sensors with good glucose responsive ability and high linearity within large detection ranges is still challenging in the field of continuous glucose detection.

Due to the characteristics of enzyme-free glucose sensors prepared from glucose-binding groups of biomaterials [19], including biocompatible, wear-resistant, and easy-to-adjust spatial network structures, such glucose sensors have strong potential for the development of implantable CGM devices. Various glucose-binding groups including natural molecules (Con A), synthetic molecules (phenylboronic acid [20,21,22], and polyacrylamide-based molecularly imprinted hydrogels) can be used to prepare enzyme-free glucose sensors [23,24,25].

Phenylboronic acid (PBA) is a chemically synthesized material that has a reversible affinity for compounds containing adjacent hydroxyl groups [26], so PBA-based hydrogels are affected by most sugars. Con A is a glucose-binding lectin isolated from jack bean (Canavalia ensiformis), which exhibits strong reversible affinity for the unmodified pyranose sugar ring at the C-3, C-4, and C-6 positions [27]. In comparison with other non-enzymatic sensors, Con-A-based hydrogel sensors have better biocompatibility and stronger specificity to glucose [28]. Yin et al. demonstrated that Con A-DexG microgel was not cytotoxic in vitro by measuring the cytotoxicity of Con A-DexG microgel on human dermal fibroblasts using a cell viability assay kit (CCK-8) [28]. Up to now, Con-A-based hydrogels in the form of hollow fibers and thin films have been developed into optical glucose sensors [29,30]. However, integrating high sensitivity, wide measuring range and low-cost fabrication into Con-A-based hydrogel sensors is still a challenge.

Flexible electronic manufacturing technology has a good application potential in the manufacture of CGM equipment, and it also provides considerable technical support for the design and manufacture of commercial CGM equipment. Glucose sensors based on flexible electronic manufacturing technology have attracted much attention in glucose concentration monitoring due to its advantages of low cost, good stability, and high sensitivity. However, there are still some challenges in modifying sensing elements (e.g., enzymes, precious metals, transition metals, etc.) on flexible substrates.

To effectively develop a flexible non-enzyme glucose sensor using flexible electronic manufacturing technology is a key research problem in the field of glucose sensor. At present, the mainstream flexible electronic manufacturing technologies include screen printing, inkjet printing, laser direct writing, and so on. Compared with other flexible electronic manufacturing technologies, laser direct writing has attracted much attention because of its advantages, such as simplicity, high efficiency and speed, low cost, and the ability to manufacture conductive layers on various substrate surfaces. Due to the advantages of laser direct-writing technology and some characteristics of lasers, it has a good potential in the development of flexible glucose sensors for the new generation of CGM equipment.

In this work, we proposed a silver-doped hydrogel sensor based on laser direct writing. After combining a Con-A-based glucose-responsive hydrogel with green-synthetic silver particles (NPs), a flexible enzyme-free glucose sensor was prepared on a laser direct-writing graphene electrode. The addition of silver particles may steady the binding of glucose to Con A, thereby improving the linearity of the sensor. The surface morphology and chemical composition of the silver NPs doped hydrogel are characterized by scanning electron microscopy (SEM), ultraviolet-visible absorption spectroscopy (UV-vis), and X-ray diffraction spectroscopy (XRD). The synthesis time of silver NPs is optimized to obtain a formula with strong sensitivity to glucose. Finally, the glucose detection capacity of the sensor is measured by the test instrument, and the relevant performance tests, such as selectivity and stability, are performed. The obtained glucose sensor maintains a sensitivity of 150.12 Ω/mM with high linearity of R^2^ = 0.97 in a large detection range of 0–30 mM glucose concentration. It has good potential in the development of CGM devices.

## 2. Materials and Methods

### 2.1. Material

Concanavalin A (Con A, extracted from jack bean, pre-activated, M_w_ = 102 kDa, containing Ca^2+^ and Mn^2+^) was purchased from Medicago Inc. (Quebec, Quebec City, Canada). Polyethylene glycol (600) diamethacrylate (PEGDMA) was obtained from TCI, Inc. (Portland, OR, USA). Dextran (M_w_ = 70kDa), glycidyl methacrylate (GMA), lithium phenyl-2,4,6-trimethylbenzoylphosphinate (LAP), d-fructose were purchased from Sigma-Aldrich (Milwaukee, WI, USA). Lactose anhydrous, maltose, uric acid (UA), and L-ascorbic acid (AA) were purchased from Macklin Biochemical Technology Co., Ltd (Shanghai, China). D-galactose was purchased from SeaSkyBio. Technology Co., Ltd. (Beijing, China), and glucose was purchased from Zhonghe Chemical Co., Ltd (Changzhou, China). Starch was purchased from Beijing InnoChem Science & Technology Co., Ltd (Beijing, China). AgNO_3_ was purchased from Sinopharm Chemical Reagent Co., Ltd (Shanghai, China). All of the regents were used as received. Kapton polyimide was purchased from Shenzhen Changdasheng Electronic Co., Ltd. (Shenzhen, China). DexG was synthesized by the ring-opening reaction of GMA and dextran based on our previous studies [31]. The obtained derivative with 20.6% substitution (DexG T70–20%) was used in the following experiments.

### 2.2. Fabrication of Graphene Electrode by Laser Direct Writing

First, the polyimide film (PI) is cleaned with alcohol and dried for later use. The designed pattern is imported into the computer software of the laser engraving machine. The laser power is set to 4200 mW, and the scanning rate is set to 6000 mm/min. Then we cut the polyimide film to an appropriate size and place it in the working area of the laser engraving machine and fix it with transparent adhesive cloth. A sheet of PI film with interfinger electrodes on the surface can be obtained by laser processing. The laser processed PI film is put into the oxygen plasma cleaning machine for surface treatment. This step can generate active groups on the surface of the PI membrane, improve surface hydrophilicity, and facilitate the adhesion of gel precursor solution and the formation of uniform film. Schematic diagram of graphene electrode by laser direct writing is shown in Figure 1a.

### 2.3. Fabrication of Con-A-Based Silver Composites

First, we added 1.0 g of starch to a beaker and slowly added 95 mL of deionized water to it. Then, we placed the beaker on a heated magnetic stirrer and placed the stirrer into the beaker. After inserting the temperature-controlled probe of the heated magnetic stirrer into the solution, we heated the solution to 40 °C and obtained a transparent and clear solution. After that, 10 mL of 0.07 M glucose solution was added to the starch solution, and then 5 mL of 1 M AgNO_3_ aqueous solution was added with continuous stirring. Subsequently, the solution temperature in the beaker was increased to 70 °C and reacted for several hours. After 3, 6, and 9 h, the appropriate amount of solution was taken out to obtain a solution containing silver particles of different particle sizes.

Next, we placed 1 mL of the solution in a beaker and added 100 mg of DexG. The beaker was then placed in an ultrasonic cleaner and shaken for 2 min to accelerate the dissolution of DexG. Then, we added 23 mg of Con A to the solution and repeated the steps of ultrasonic concussion described above. Finally, 10 wt% of PEGDMA and 1 wt% of the photoinitiator LAP were added, and the ultrasonic concussion was performed again for 2 min to obtain a gel precursor solution containing silver particles that were reduced for 3, 6, and 9 h.

### 2.4. Fabrication of the Silver-Doped Glucose Sensor

The glucose-responsive hydrogel was formed in situ on the ICEs. To begin with, 10 μL of gel precursor solution was dropped onto the treated interdigital electrode to ensure that the solution penetrated the interdigital electrode area. Then, the hydrogel was formed either by UV curing or laser direct writing. For UV curing, the sample was subjected to ultrasonic vibration at room temperature for 2 min. For laser solidification, the PI film was placed in the working area of the laser engraving machine and engraving was started. The laser was scanned in the gel precursor solution to solidify the hydrogel. The laser intensity was 106 mW. After curing, a thin layer of hydrogel was formed on the interdigital electrode, which was consistent with the size of the interdigital boundary. The prepared sensor was further immersed in deionized water to store the hydrogel sensor so as to maintain its response to glucose. In order to simplify the writing, the hydrogel sensors formed by silver particles solution with different reduction times of 3, 6 and 9 h, are named Ag^3^/DexG-ConA sensor, Ag^6^/DexG-ConA sensor, and Ag^9^/DexG-ConA sensor, respectively.

### 2.5. Characterization of the Con-A-Based Silver Composite

A scanning electron microscope (SEM) was used to observe the morphology of hydrogel-lyophilized samples with Hitachi S-3400 equipment (Tokyo, Japan). The absorption of ultraviolet and visible light of Con-A-based silver composites was detected by ultraviolet and visible absorption spectroscopy (UV-vis) to determine the loading of Ag particles in hydrogels. IM3536 LCR METER HIOKI (IM3536 LCR METER HIOKI) was used to monitor the resistance change of the sensor. It was connected to a PC through a USB cable. On the PC, the corresponding software was used to record the resistance change of the sensor with the concentration. The device was set to a voltage of 5 V and a frequency of 30 KHz. The device was set to a voltage of 5 V and a frequency of 30 KHz. The detection device of the LCR tester was respectively connected to both ends of the interdigital electrodes and recorded them according to the value displayed by the software.

### 2.6. Performance of the Silver-Doped Glucose Sensor

The resistance of the silver-doped glucose sensor was investigated by a ST2253 four-point probe meter (Suzhou Jingge Electronic Co., Ltd., Jiangsu, Suzhou, China). The resistance of the laser-carved graphene electrode at a glucose concentration of 0–30 mM was tested to determine whether it interfered with glucose monitoring. The external voltage was 220 V, and the current range was selected automatically. In the same environment, the resistance data of Ag^3^/DexG-Con A, Ag^6^/DexG-Con A and Ag^9^/DexG-Con A hydrogel sensors were analyzed, and the hydrogel sensor with the highest sensitivity was selected for further performance research. The response time of Ag^6^/DexG-ConA hydrogel sensor was studied in 5mM glucose solution and compared with the glucose concentration response of hydrogel (DexG-Con A) without silver composite. The response of the hydrogel sensor was compared between laser direct writing and UV light molding at 0–30 mm glucose level. The hydrogel glucose sensor was alternately put into 0 and 5 mM glucose concentration solution to test the stability of the sensor.

## 3. Results and Discussion

### 3.1. Principle of the Proposed Silver-Doped Hydrogel Sensor

The silver-doped hydrogel sensor consists of the Ag/DexG-Con A hydrogel and interdigital carbon electrodes (ICEs) formed by laser direct writing. The DexG-Con A hydrogel has good biocompatibility and rapid responsibility and easily adjusts the network structure [31]. The DexG-Con A hydrogel is a glucose sensitive hydrogel, which has a dual-network structure. The covalent bonds among the DexG molecules provide mechanical support for the hydrogels, and the specific physical binding between Con A and DexG is conducive to the glucose responsiveness [28]. The competitive binding between DexG-Con A and glucose (Glu)-Con A affects the structure of the DexG-Con A network, resulting in changes in the volume and permittivity of the hydrogel. Additionally, the binding process can be represented by Equations (1) and (2), according to the ligand competition theory [28,32]: (1)$$\left(\mathrm{Dex}|\mathrm{GMA}\right)+\mathrm{Con}\;A↔\langle \mathrm{Dex}|\mathrm{GMA}|\mathrm{Con}\;A\rangle ,$$
(2)$$\mathrm{Glucose}+\mathrm{Con}\;A↔\left(\mathrm{Glucose}|\mathrm{Con}\;A\right),$$

Figure 1 exhibits the volume change principle of Ag-NP-doped DexG-Con A hydrogel. As shown in Figure 1c, when the glucose concentration increases, the Con A molecules preferentially bind with glucose rather than DexG, leading to the equilibrium of Equation (2) shifting towards the right side. At the same time, the equilibrium of Equation (1) shifts towards the left. This causes the expansion of the DexG-Con A network and thus the swelling and permittivity change of the hydrogel. Contrarily, the hydrogel volume shrinks when the glucose concentration decreases.

### 3.2. Characterization of the Con-A-Based Silver Composites

The Ag^3^/DexG-Con A, Ag^6^/DexG-Con A, and Ag^9^/DexG-Con A composites were characterized by SEM and EDS techniques, and the microstructure characteristics of different composites could be observed. Figure 2 shows the surface SEM and cross-section SEM of freeze-dried hydrogels of Ag^3^/DexG-Con A, Ag^6^/DexG-Con A, and Ag^9^/DexG-Con A composites. The composites samples have similar rough surface covered with silver NPs as well as porous cross-section structure owing to the network of the DexG-Con A hydrogel [33]. Figure 3 exhibits the Ag mapping of different composites by EDS analysis. It can be seen that the distribution of Ag element is uniform both on the surface and cross-section of the freeze-dried gel samples.

By further analysis of the EDS data, Figure 4a shows the trend of Ag weight percentage (wt%) under different reaction times. On the surface of the freeze-dried hydrogel loaded with silver particles, the Ag content decreased with the increase in reaction time. According to the cross-section data, the Ag content increased first and then decreased. It may be because that the Ag particles with a long reaction time have a smaller particle size [34,35]. Therefore, Ag^3^ NPs may be mostly distributed on the surface of the hydrogel due to the large NP size, while Ag^9^ NPs are prone to be penetrated into as well as leaked from the hydrogel. Considering the above results, we can preliminarily infer that the Ag^6^/DexG-Con A hydrogel may have better sensing performance.

Figure 4b shows the UV-Vis absorption spectra of freeze-dried Ag^3^/DexG-Con A, Ag^6^/DexG-Con A and Ag^9^/DexG-Con A hydrogels. All three samples have obvious absorption peak at about 430 nm, which is the plasma resonance absorption characteristic peak of silver particles [36]. The Ag^6^/DexG-Con A hydrogel has the highest light absorption, indicating the Ag content in the Ag^6^/DexG-Con A hydrogel may be the highest, which is in consistent with the results obtained from EDS. Figure 4c shows the XRD patterns of the three hydrogel samples. The peaks at 2θ = 38.74°, 42.2°, 65.12°, and 77.32°correspond to the crystal faces of (111), (200), (220), and (311), respectively [37]. XRD results confirm that silver NPs were successfully synthesized after chemical reaction. Figure 4d shows the change of square resistance of the hydrogel samples with or without Ag NPs. After doping Ag NPs into the hydrogel, the resistance dramatically decreased to a certain extent, indicating that the incorporation of Ag NPs can indeed enhance the conductivity of the hydrogel. 

In addition, compared with Ag^3^/DexG-Con A and Ag^6^/DexG-Con A hydrogels, the Ag^9^/DexG-Con A sample shows inconspicuous resistance change in response to different glucose concentrations. So, the Ag^9^/DexG-Con A hydrogels have worse sensing performance. Compared with Ag^3^/DexG-Con A hydrogels, the Ag^6^/DexG-Con A sample shows lower resistivity, and when the glucose concentration changes, the resistance of Ag^6^/DexG-Con A hydrogel changes more, indicating that it is more sensitive. Therefore, of the three hydrogels, the Ag^6^/DexG-Con A hydrogels have better sensing performance.

### 3.3. Performance of the Silver-Doped Hydrogel Sensor

The resistance of laser engraving graphene electrode [38] at 0–30 mM glucose concentration was detected to eliminate the interference of laser engraving graphene electrode on glucose monitoring. As shown in Figure 5a, with the increase in glucose concentration, the measured resistance signal changes are disorderly, and the values are basically located near 46,000 Ω. This indicates that the pure laser engraving graphene electrode does not have the ability to detect glucose, and the influence of the electrode on the detection of glucose by polymer-based glucose sensors can be excluded. Figure 5b shows that the resistance of Ag^3^/DexG-Con A, Ag^6^/DexG-Con A, and Ag^9^/DexG-Con A hydrogel sensors increase with the increase in glucose concentration, and the resistance value is basically near 23,000 Ω. This is quite different from the resistance value and resistance change of the above laser engraving graphene electrodes in the glucose concentration test, which also indicates that the laser engraving graphene electrode has little effect on glucose detection.

By analyzing the resistance data of hydrogel sensors, the response ability of different hydrogel sensors to glucose can also be obtained, as shown in Figure 5b [39]. The sensitivity of Ag^3^/DexG-Con A, Ag^6^/DexG-Con A, and Ag^9^/DexG-Con A hydrogel sensors were 78.75 Ω/mM (R^2^ = 0.97), 150.12 Ω/mM (R^2^ = 0.97), and 84.20 Ω/mM (R^2^ = 0.96), respectively. It can be seen from the calculation data that the sensitivity of the hydrogel sensor increases first and then decreases, indicating that the performance of the Ag^6^/DexG-Con A hydrogel sensor is the best, which is also consistent with the characterization results of the aforementioned EDS, UV-Vis absorption, and square resistance. Therefore, in the following experiments, the Ag^6^/DexG-Con A hydrogel sensor is selected for further performance tests. Most importantly, all three samples doped with Ag NPs show high linearity of R^2^ > 0.95 within the glucose range of 0–30 mM. Compared with Cai’s work [31] with R^2^ = 0.54 and Yin’s work with R^2^ = 0.84 [40], the proposed sensor has the highest linearity within the large glucose concentration range of 0–30 mM among the reported hydrogel-based glucose sensors. This may be due to the doped Ag NPs filling part of the pores of the hydrogel, limiting the diffusion rate of glucose molecules in the hydrogel, thus improving the linear reliability of the hydrogel sensor. The response time of Ag^6^/DexG-Con A hydrogel sensor to 5mM glucose, as shown in Figure 5c, demonstrates that it takes about 12 min for the hydrogel sensor to reach equilibrium resistance response at a glucose concentration of 5 mM, which is six times longer than DexG-Con A hydrogel sensor without Ag NPs doped in Cai’s work [31]. This result confirms the possible reason for the aforementioned high linearity.

Figure 5d shows that the resistance response of Ag^6^/DexG-Con A hydrogel sensor prepared by directly solidifying the hydrogel during the laser direct-writing process exhibits similar performance as that sensor with UV-light-cured hydrogel. The resistance increases with the increase in glucose concentration, and the sensitivity of the sensor is calculated to be 169.0 Ω/mM (R^2^ = 0.932), with no significant difference compared with the Ag^6^/DexG-Con A hydrogel sensor in Figure 5b (150.12 Ω/mM, R^2^ = 0.97). These results indicate the great potential of laser direct writing in hydrogel sensor fabrication; therefore, more sensitive ability of hydrogel sensors may be formed by the easy structural design of hydrogels using laser direct writing. Furthermore, the reversible response to glucose is a great advantage of both PBA-based and Con-A-based hydrogel sensors [30,39,41,42]. In order to investigate the reversible glucose response of the Ag-NP-doped DexG-Con A hydrogel sensors, the capacitance in response to the glucose solution with repeated concentration change between 0 and 5 mM was recorded, and the results are shown in Figure 5e. The average resistance response of the sensor at 0 mM glucose concentration was 1.000, and the standard deviation was 0.003. The mean resistance response of the sensor at 5 mM glucose concentration was 1.023 s and the standard deviation was 0.005. These results show that the sensor has good stability and reversible response to glucose. It is also proved that Ag^6^/DexG-Con A hydrogel glucose sensor can be reused [40].

Selectivity is an important characteristic to evaluate the performance of glucose sensors. The sensor is interfered with by other sugars (including galactose, fructose, lactose, and sorbitol) in the interstitial fluid during blood glucose detection. The physiological concentration of these disturbances is about an order of magnitude lower than the normal glucose level. In response to disturbances in the 5 mM glucose buffer solution, the resistance of the Ag^6^/DexG-Con A hydrogel sensors with their concentration of ten percent glucose was measured (at 30 kHz). The experimental result is shown in Figure 5f. The resistance ratio is used to indicate the glucose response in the presence of potential disturbances, which is calculated according to R_d_/R_g_, where R_g_ is the resistance of the glucose solution, and R_d_ is the resistance of the glucose solution to different disturbances. In response to the existence of galactose, fructose, lactate, and sorbitol, the obtained resistance ratios are 1.13, 1.13, 1.13, and 1.17, respectively. This demonstrates that the Ag^6^/DexG-Con A hydrogel sensor is resistant to the disturbances of other sugars in the interstitial fluid. In order to avoid other ions in the interstitial fluid affecting glucose detection, the sensor can be integrated with an ion-selective permeable membrane in the future. The Ag^6^/DexG-Con A hydrogel sensor is promising for implantable devices.

## 4. Conclusions

In this study, we designed a Con-A-based silver-doped hydrogel sensor that can be used for wearable devices and continuous glucose monitoring (Appendix A). A flexible enzyme-free glucose sensor was fabricated by combining Con-A-based glucose-responsive hydrogel with green-synthesized silver particles on laser-written graphene electrode by laser direct-writing technology. The experimental results showed that in a glucose concentration range of 0–30 mM, the proposed sensor is capable of measuring the glucose level in a repeatable and reversible manner, showing a sensitivity of 150.12 Ω/mM with high linearity of R^2^ = 0.97. Compared with the sensor hydrogel formed by UV light curing, the hydrogel made by laser direct writing was successfully prepared in this paper. It has similar performance to the common hydrogel sensor formed by UV light, and it is also expected to improve the response ability of the sensor by designing the structure of hydrogel. On the premise of compact structure, this sensor solves the problem of insufficient linearity of hydrogel-based glucose sensors within a large detection range (0–30 mM) and improves the application potential of hydrogel-based glucose sensors in CGM device development.

## Figures and Tables

**Figure 1 polymers-15-01423-f001:**
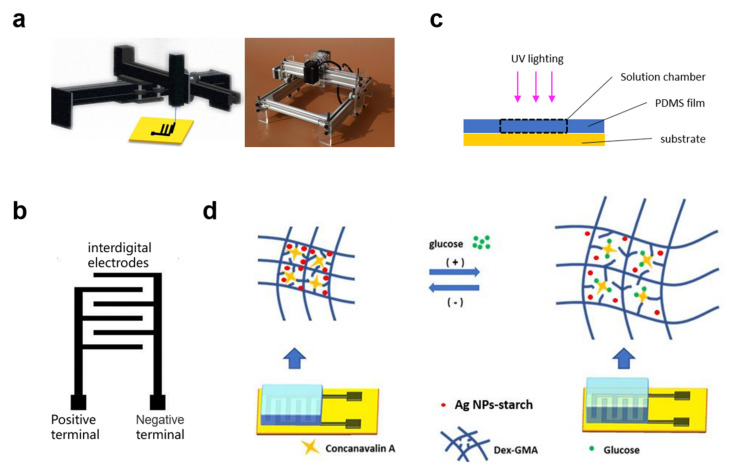
(**a**) Laser direct writing to form interdigital electrodes on PI film; (**b**) pattern of interdigital electrodes; (**c**) illustration of hydrogel formation through UV lighting; (**d**) illustration of glucose-sensing principle: glucose binding and hydrogel swelling process.

**Figure 2 polymers-15-01423-f002:**
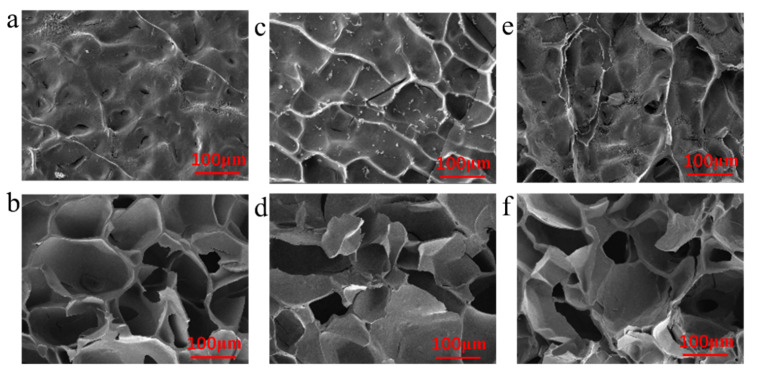
(**a**,**b**) Surface SEM and cross-section SEM of freeze-dried Ag^3^/DexG-Con A hydrogel samples. (**c**,**d**) Surface SEM and section SEM of freeze-dried Ag^6^/DexG-Con A hydrogel samples. (**e**,**f**) Surface SEM and cross-section SEM of freeze-driedAg^9^/DexG-Con A hydrogel samples.

**Figure 3 polymers-15-01423-f003:**
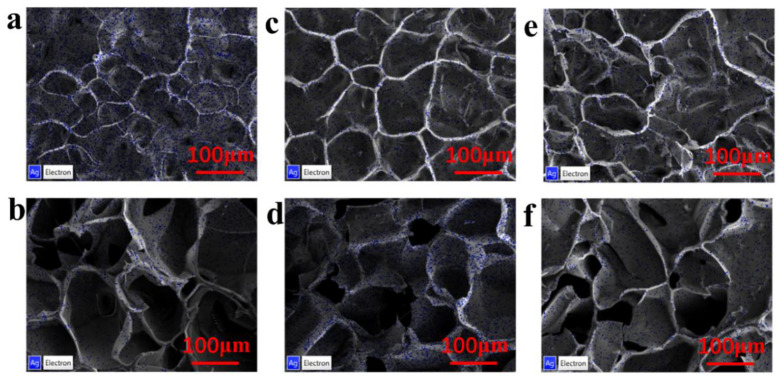
(**a**,**b**) EDS of Ag on surface and cross-section of freeze-dried Ag^3^/DexG-Con A hydrogel. (**c**,**d**) EDS of Ag on surface and cross-section of Ag^6^/DexG-Con A hydrogel freeze-dried samples. (**e**,**f**) EDS of Ag element on surface and cross-section of freeze-dried Ag^9^/DexG-Con A hydrogel.

**Figure 4 polymers-15-01423-f004:**
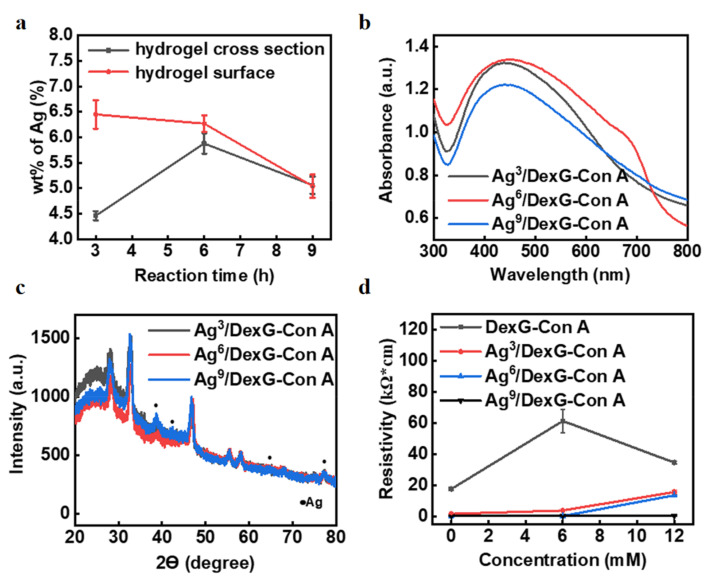
(**a**) The weight percentage (wt%) of Ag element in surface and cross-section of hydrogel loaded with Ag NPs under different reaction times. (**b**) Uv-vis absorption spectra of Ag^3^/DexG-Con A, Ag^6^/DexG-Con A and Ag^9^/DexG-Con A hydrogel samples. (**c**) XRD spectra of Ag^3^/DexG-Con A, Ag^6^/DexG-Con A and Ag^9^/DexG-Con A hydrogels. (**d**) Square resistance diagrams of Ag^3^/DexG-con A, Ag^6^/DexG-con A and Ag^9^/DexG-con A hydrogels.

**Figure 5 polymers-15-01423-f005:**
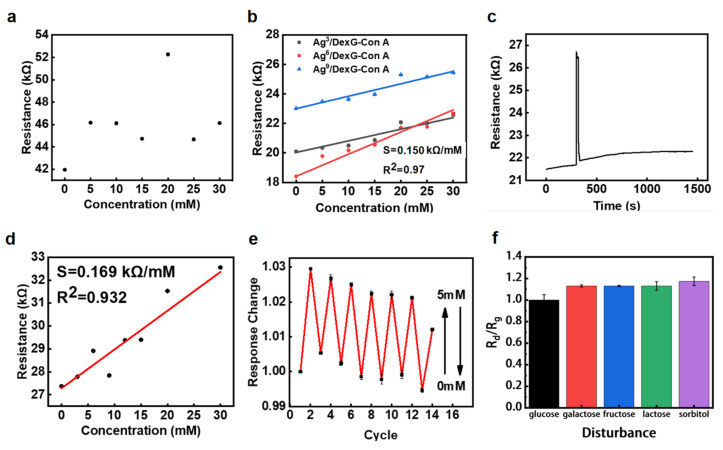
Resistance of proposed sensor from multiple measurements. (**a**) Relationship between resistance change and glucose concentration of laser-carved graphene electrode. (**b**) Resistance response of Ag^3^/DexG-con A, Ag^6^/DexG-Con A, and Ag^9^/DexG-Con A hydrogel sensors prepared by UV lighting to glucose. (**c**) Resistance response of Ag^6^/DexG-Con A hydrogel sensor to 5mM glucose. (**d**) Resistance response of Ag^6^/DexG-Con A hydrogel glucose sensor prepared by laser direct writing under different glucose concentrations. (**e**) The resistance response of Ag^6^/DexG-Con A hydrogel glucose sensor was changed by placing it alternately in 0 and 5 mM glucose concentration solution. (**f**) Resistance in response to disturbances in 5 mM glucose buffer solution.

## Data Availability

https://pan.baidu.com/s/1Tz3srFTytCbJn1T6FO9YMg?pwd=eao3, accessed on 9 March 2023.

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
