# Peer review of "A High-Linearity Glucose Sensor Based on Silver-Doped Con A Hydrogel and Laser Direct Writing"

_polymers, 2023, doi:10.3390/polym15061423_

Round 1
Reviewer 1 Report
• The main question was the development of a glucose sensor with a high sensitivity, wide measurement range and low cost was proposed using a hydrogel based on Concavalin A. • The topic is relevant, due the need of a Continuos Glucose Monitoring System. Till now, the development of flexible glucose sensors with good glucose responsive ability and high linearity within large detection range is still challenging in the field of continuous glucose detection.• High linearity of R2 >0.95 within the glucose range of 0-30 mM. Compared with other works with R2 =0.54 and R2 =0.84, the proposed sensor has the highest linearity within the large glucose concentration range of 0-30 mM among the reported hydrogel-based glucose sensors.
• The research was well managed, the tests and characterization carried out on the sensor are adequate. • Tests and characterization carried out on the sensor show that the proposed proposal was satisfactorily fulfilled. • Figures are well discussed and no more are needed.
In Line 41. What means BG?
In line 160, What is the purpose of use XPS test?
Reviewer 2 Report
This can be improved in the views of the comments below.
1. A good schematic of the work will improve the quality of the work
2. Section 2.2 should not be written as direction but it should be written in a way that shows how the authors did it.
3. How advantageous is the laser writing to spin coating or drop coating etc. methods
4. As mentioned. no XPS data were shown
5. Real sample should be run and more realistic interference species should be used to investigate the resistance (fig 5f)
6. More discussion about glucose sensing such as non-enzymatic sensors is needed in the introduction such as in Talanta, 2016, 149, 30-42
Reviewer 3 Report
The authors presented the continous glucose monitoring on a graphene electrode. Nevertheless the manuscript content is not well described and lead to unclear method or platform that has been use in this experimental.
1. The illustration of the platform, fabrication and its method should be presented for a better clarity. How the graphene electrode is obtained, how to add the Ag/DexG/ConA on the top of graphene electrode, this illustration will be a goo way to summary the whole method section. Hhow the measurement setup? Especially the signal that were recorded is resistance, instead of current or voltage. In Fig. 1, the graphene legend should be added.
2. Why the resistivity trend in the graphene electrode Fig. 5 (a) is not in the trend, what is the reason and the mechanism behind. While in the proposed sensing (Fig. 5.b) it has trend there. The fundamental different between only graphene and proposed electrode should be well discussed. Only mention Fg. 5 a, is like this trend and Fig. 5 b is like that trend, is not a good discussion section.
3. The R^2 in Fig. 5 d. is not good enough from statistical point of view. The minimum value of 95% of R^2 is usually the threshold of the data trend in the scientific article.
4. The authors claim the system is continues glucose monitoring, however, there is only realtime signal for a single concentration. The series concentration of measurements in realtime should be presented as well, to see the "continuous" performance.
Based on the review above, the scientific content of the article is not recommended for publication in Polymer.
Round 2
Reviewer 2 Report
The authors have improved the quality of their work substantially. However, real sample analysis is not being done to avoid ethical issues. In this case, authors could use simulated serum or urine for the glucose.
Reviewer 3 Report
The revised manuscript has been improved in terms of clarity and scientific content. The authors addressed the reviewer's comments with the revision in the main manuscript and the additional data. The revised manuscript can be considered for publication in Polymers.
